# Research on Static Stability of Firefighting Adapter

Jaroslav Matej [1,*] , Richard Hnilica [2] and Michaela Hnilicová [1]

1 Department of Mechanics, Mechanical Engineering and Design, Faculty of Technology, Technical University in Zvolen, T.G. Masaryka 24, 96001 Zvolen, Slovakia; michaela.hnilicova@gmail.com
2 Department of Manufacturing Technology and Quality Management, Faculty of Technology, Technical University in Zvolen, T.G. Masaryka 24, 96001 Zvolen, Slovakia; hnilica@tuzvo.sk
* Correspondence: jaroslav.matej@tuzvo.sk

**Abstract:** The article is focused on the static stability of a vehicle with a 2000-liter water tank behind the rear axle. The purpose of the research is to evaluate the influence of the tank on stability. The vehicle is composed of a forestry skidder, a water tank, and equipment. To equilibrate the tank a ballast weight of 500 kg in front of the skidder is used. The influence of various combinations of the full water tank, half-filled water tank, and ballast weight is evaluated. The stability is determined as the distances of vectors of a stability triangle and a gravity vector, that is placed in a center of gravity of the whole vehicle. A Python programming language is used to implement the solution. The results are displayed in form of heatmaps. For the analyses, a slope angle of 18° is used. The results show that the longitudinal stability is decreased and lateral stability is increased. The tank also makes the vehicle unstable in some positions.

**Keywords:** stability triangle; forest fires; firefighting vehicle; forest wheeled skidder; rear shield; water tank





## 1. Introduction

The changing climate with warmer, drier conditions is likely to increase the risk, timing, and severity of forest fires [1]. A fight against the fires then will require increased effort, new solutions, or at least more solutions. Elimination of forest fires on any scale combines several phenomena. Firstly, sufficient forces and resources for its localization and liquidation. Secondly, the availability of the site and a sufficient extinguishing agent (water) in connection with the availability of water resources themselves. Not one area of this vicious circle cannot be prepared in advance. However, making the forest accessible with the preservation, revitalization and construction of the forest road network and water resources should be part of the construction of the forest [2]. It is necessary to look at safety construction from the point of view that it is not just a question of designing and dimensioning machinery, but the creation of a comprehensive system of filling defined objectives. The task of designers who deal with technical (operational) parameters is to ensure the highest possible efficiency of the machine [3,4]. At present, there are many vehicles, specially designed or modified, to fight wildfires. They are mainly one-purpose machines [5–9]. The key point is the need to transport water close to the nearest intervening fire brigade, as soon as possible. Then, the most appropriate solution, for forest fires, is the use of ordinary forest machinery. Forestry skidders provide high slope accessibility and workability in forest terrain. If they are equipped with firefighting accessories, especially without the need to intervene in the construction of the skidders, the time of liquidation of a forest fire can be shorter. However, it reduces the possibility to tune up the parameters of such a firefighting system. The firefighting adapter (Figure 1) allows the transport of water into an area of fire when connected to the forestry skidder. Based on the field conditions of forests and the available technology, it can be stated that the construction of the adapter is a new solution for the liquidation of forest fires in difficult forest conditions [10]. Its

essential feature is a big water tank placed far beyond the rear axle, on a rear shield. This solution does not intervene in the construction of the skidder. However, the big tank has a big influence on stability. The research of stability herein proposed is inspired by this innovative design of the firefighting adapter [10]. The adapter with a 2000 L water tank (Figure 1) is attached to the forest skidder LKT-81T. The water tank can be filled with water repeatedly, after its final placement in a place accessible from a wildfire area, using a helicopter equipped with a Bambi bucket. From a firefighting point of view, the bigger tank, the better tank, however the bigger tank behind the rear axle, the bigger influence on stability. In general, a big tank in the rear part can be equilibrated with a ballast weight in the front part. The purpose of this study is to investigate such a vehicle from point of view of static stability.

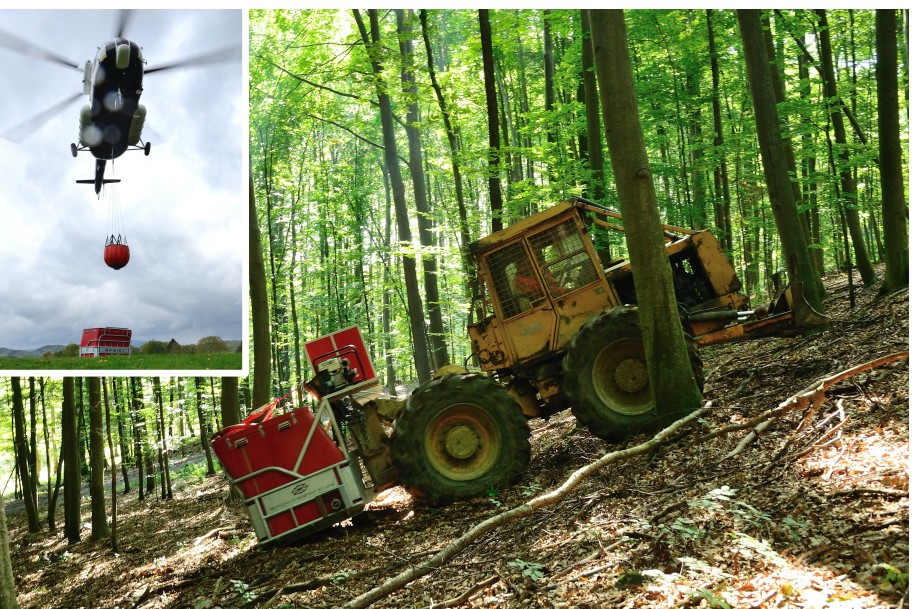

**Figure 1.** Firefighting adapter attached to forestry skidder. Filling water tank with Bambi bucket.

Many methods have been proposed to solve static stability. They include various types of machines equipped with front/rear axle pivot joints and/or articulated frames and/or using a stability triangle and/or similar computational methods to solve the static stability of a vehicle. Research of the firefighting adapter [10] contains no stability analysis. In addition, the solution is quite specific. However, the physical principles and methods, usable for the solution of stability, are valid for all vehicles. Gibson [11] performed research on the side slope stability of articulated-frame logging tractors. He defined also a stability triangle and the center-of-gravities locations as approximations. Franceschetti [12] focused on the lateral stability performance of articulated narrow-track tractors. He used an inclined plane, also for an articulated tractor. Sierzputowski [13] used an inclined plane platform to measure the rollover stability of an articulated vehicle, and noted that theoretically, for the complete assessment of the vehicle rollover stability, the critical slope angles should be measured at all the angular orientations of the platform. Bołoz [14] investigated the stability of articulated drilling rigs. He created a mathematical model of the rig with a global coordinate system fixed to the front frame. Tomašić [15] measured the distribution of forces during uphill and downhill timber skidding at various slope angles. Bietresato and Mazzetto [16–18] created a tiltable plane/platform, as a wide flat structure (15 × 15 m), able to simulate different slope angles, allowing a vehicle to maneuver/travel on it along circular paths in a controlled and safe environment. For static tests, they used a turntable inserted in the lower half-platform. It allows testing a vehicle's stability in all the orientations on the inclined plane. Majdan [19] measured the position of the center of gravity of a small agricultural tractor, and performed a calculation of static overturning angle,

according to ISO 16231-2. Mazzetto [20] investigated the kinematics of an articulated tractor with a central joint (2 degrees of freedom) using an application written in Matlab. Demšar [21] created a mathematical model and numerical simulation of the static stability of an agricultural tractor.

These researches and methods use similar principles to solve the static stability of a vehicle on a slope. However, if a vehicle is required to carry two or more loads in general positions, the methods above are not directly usable. It is a reason why the research method herein proposed has been developed. The method displays all the possible ranges of an articulation of the frame, and positions/rotations of the skidder on an inclined plane as a function of two independent variables, and displays them in form of a heatmap.

## 2. Materials and Methods

The firefighting adapter can be attached to the LKT-81 skidder (Figure 1). It was used in the research because it is frequently used and available in Slovakia. However, this machine is not manufactured at present. Thus, it will not be used as a base machine for the adapter in the future. This is why the LKT-81 skidder was not used directly for analyses of stability, but a simplified alternative skidder model with similar parameters (Table 1) was used. This simplification includes approximated positions of FG, and RG points (centers of gravities), placed on the centerline. The main goal of this research is to compare various configurations (Table 2) of the water tank and additional front ballast weight, and not to evaluate an individual configuration, joined with the real (LKT-81) skidder. On a real vehicle, the FG and RG points are not placed exactly on the centerline but have a certain offset (in mm) from the centerline.

**Table 1.** Basic parameters of alternative skidder model (as shown on Figure 2).

| | |
|---|---|
| Coordinates of important points (x, y, z) [mm] | FPP (−1200, 0, 500), RRW (1200, 1010, 0), RLW (1200, −1010, 0), FRW (1200, 1010, 0), FLW (1200, −1010, 0), FG (−1100, 0, 1000), RG (1100, 0, 800), FAW (−1700, 0, 340), RAW (2825, 0, 926) |
| Gravities [N] | FG (gravity) = 0.615 × 7145 kg × 9.81 m/s$^2$ = 43,106 N, RG (gravity) = 0.385 × 7145 kg × 9.81 m/s$^2$ = 26,985 N (skidder's weight of 7145 kg, where 61.5% is on the front axle) |
| Loads: | A and/or B at RAW and/or FAW positions, as described in configurations (Table 2). |

**Table 2.** Various configurations of additional loads. A—weight of water tank within RAW point, B—additional load in front mount point FAW (as shown in Figure 2), X—applied load.

| Loads A and/or B/Configuration | C1 | C2 | C3 | C4 | C5 |
|---|---|---|---|---|---|
| B = 0 | X | X | X | | |
| A = 0 | X | | | | |
| B = 500 kg | | | | X | X |
| A = 2688 kg (2000 L) | | X | | X | |
| A = 1688 kg (1000 L) | | | X | | X |

A method used to evaluate the stability is based on a physical principle, that says that a body/vehicle is statically stable if a gravity vector placed in a center of gravity stays inside of a stability polygon/triangle. The stability triangle is composed of tipping lines a vehicle overturns around when losing its stability. Then, the solution is to create lines of a stability triangle and a gravity vector in form of equations of spatial lines. Then, the distances of the gravity vector to the lines of the stability triangle determine the stability of the vehicle, in the purest form. Subsequently, a test of whether the gravity vector stays

inside the stability triangle is necessary, to check unstable positions. Then the results can be visualized in plots for all the sides of the stability triangle. The solution based on vectors and a general programming language provides a possibility to make the solution universal. Then, it is possible to modify it in the future for different types of machines. The solution the stability has been implemented in Python programming language in a 3-dimensional space. Important points of the skidder then have three coordinates in the local coordinate system (Figure 2).

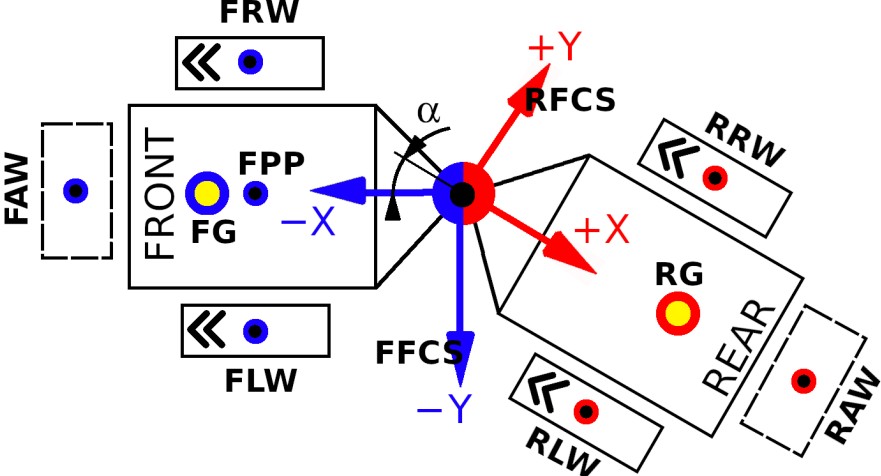

**Figure 2.** Important points of the skidder: FRW, FLW, RLW, RRW—coordinates of contact points of tires with slope/terrain, FG, RG—coordinates of centers of gravities for front and rear frames of skidder, FAW, RAW—coordinates of front and rear mount points for additional weights/loads, FFCS—front frame coordinate system (with Z-axis toward up), RFCS—rear frame coordinate system (with Z-axis toward up), FPP—coordinates of the front axle pivot point.

The stability triangle is created by lines ($l1\_z$, $l2\_z$, $l3\_z$) as shown on Figure 3. The gravity vector is placed in the center of gravity marked T. It is also represented as a line that we used to compute distances to the lines of the stability triangle.

Fully 3D equations of lines in vector form were used, as follows:

$$(x, y, z) = (x_0, y_0, z_0) + t(a, b, c) \tag{1}$$

where:

$(x_0, y_0, z_0)$—a point on a line (e.g., start point),
$(x, y, z)$—another point on a line (e.g., end point),
$t$—a parameter describing a particular point on a line,
$(a, b, c)$—directional vector of a line.

Distances between lines, mainly between the gravity vector and tipping lines of the stability triangle (Figure 3) were determined using Equation [22]:

$$d = \left| \frac{(\vec{V_1} x \vec{V_2}) \overrightarrow{P_1 P_2}}{\left| \vec{V_1} x \vec{V_2} \right|} \right| = \left| \frac{(q_1 r_2 - q_2 r_1) a_{12} + (r_1 p_2 - r_2 p_1) b_{12} + (p_1 q_2 - p_2 q_1) c_{12}}{\sqrt{(q_1 r_2 - q_2 r_1)^2 + (r_1 p_2 - r_2 p_1)^2 + (p_1 q_2 - p_2 q_1)^2}} \right| \tag{2}$$

where:

$p_1 = x1 - x1_0; q_1 = y1 - y1_0; r_1 = z1 - z1_0$
$p_2 = x2 - x2_0; q_2 = y2 - y2_0; r_2 = z2 - z2_0$
$a_{12} = x1_0 - x2_0; b_{12} = y1_0 - y2_0; c_{12} = z1_0 - z2_0$

and

$x1, y1, z1$—are $x, y, z$ coordinates of an end point of the first line (see Equation (1))

$x2, y2, z2$—are $x, y, z$ coordinates of an end point of the second line
$x1_0, y1_0, z1_0$—are $x, y, z$ coordinates of a start point of the first line
$x2_0, y2_0, z2_0$—are $x, y, z$ coordinates of a start point of the second line

However, distances computed using Equation (2) are always positive. That is why the crossing number algorithm was implemented to detect stable or unstable positions of the skidder. It says that if a point is on the outside of the polygon, the ray from the point in any fixed direction will intersect its edge an even number of times. An odd result means the point is on the inside of the polygon.

Equation (2) needs to know position T of the gravity vector (Figure 3). It is computed from weights and gravity vectors positions of the front and rear frames of the skidder (FG, RG, see Figure 3) and mount points for additional weights/loads (FAW, RAW, see Figure 3). All the gravity vectors create a list. The procedure for getting the final gravity vector position T is as follows:

1. Remove two vectors from the list, and combine them into one vector;
2. Inserted the vector into the list;
3. Repeat if the list contains more than one vector;
4. The last vector is the final gravity vector T.

Equations (1) and (2) require following input parameters:

- Coordinates [x, y, z] [mm].
    - Coordinates of wheel's contact points with a terrain (RRW, RLW, FRW, FLW) (Figures 2 and 3);
    - Coordinates of a front axle pivot point FPP;
    - Coordinates of centers of gravities for front (FG) and rear (RG) frames;
    - Coordinates of centers of gravities of additional loads (FAW, RAW).
- Gravities [N].
    - Gravities (FG, RG, FAW, RAW).

Positions of FRW, FLW (front wheels) points and loads (FAW and/or RAW) that are not applied (Figures 2 and 3) are not mandatory. Equation (2) computes stability for one skidder's position on a slope. To obtain complex information on static stability the computations were solved in a loop in two modes with different input values:

Input values for 1st mode:

- Slope angle $\beta$ is a constant;
- Frame's articulation $\alpha$ is variable in interval of $\langle -45°; 45° \rangle$;
- Skidder's rotation $\gamma$ on a slope is variable in interval of $\langle 0°; 359° \rangle$.

Input values for 2nd mode:

- Frame's articulation $\alpha$ is a constant;
- Slope angle $\beta$ is variable in interval of $\langle 0°; 45° \rangle$;
- Skidder's rotation $\gamma$ on a slope is variable in interval of $\langle 0°; 359° \rangle$.

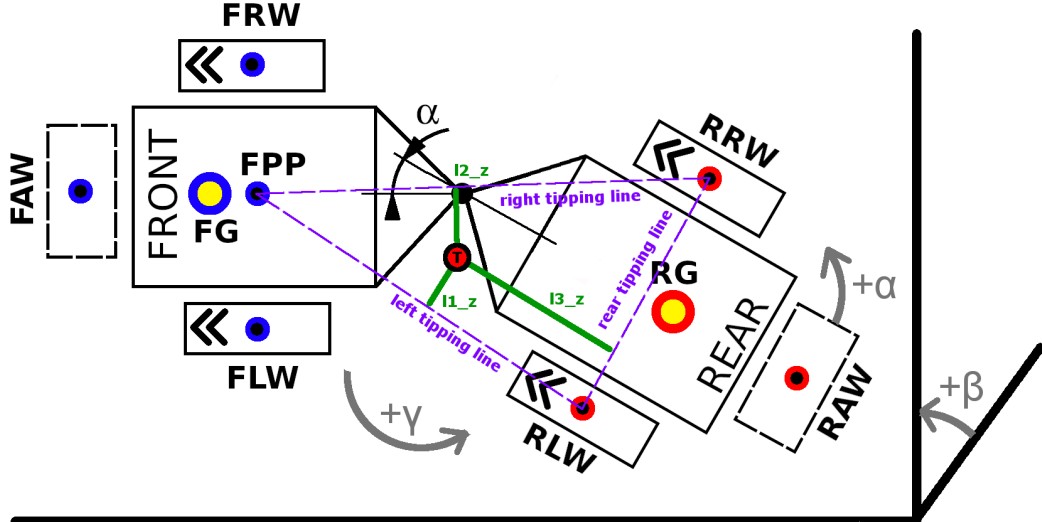

**Figure 3.** Position of skidder with front axle oscillation on slope: α—articulation of frame [°], β—slope angle [°], γ—rotation (=position) of skidder on slope [°], $l1\_z$, $l2\_z$, $l3\_z$—sides of stability triangle (left, right, and rear one). RLW, RRW—coordinates of contact points of rear tires with slope/terrain, T—center of gravity of skidder. Three green lines show distances of gravity vector (through T, center of gravity) to sides of stability triangle (=tipping lines) in 3D space.

The first mode can evaluate the static stability of a firefighting device configuration on a predefined slope, considering maneuvers, i.e., rotations/positions on the slope. Thus, it is possible to evaluate an influence of a combination of a frame articulation α and rotation γ of a firefighting device configuration on stability, as a distance of a gravity vector to a tipping line. In general, wheeled skidders should not operate on slopes that exceed 40%. It is, of course, a function of load size. For a skidder with a 3-tonne load, it is between −34% (downhill) and +39%, a skidder with a 4-tonne load between −35% and +30% and skidder with 5-tonne load between −41% and +11% [23]. The maximum load size of the firefighting adapter is 3188 kg. Considering the position of the water tank the angle of 18° (32.5%) has been used. This angle was used for the analyses in the first mode, where a slope angle β is a constant. The second mode allows to set up a constant angle of articulation of a frame and investigates its influence on stability if needed. Or it allows determining a maximal slope angle for a given combination of input parameters. Both the modes, require two variable input parameters, and provide three outputs, as shown in Figure 4 (1st mode) and Figure 5 (2nd mode). A Python matplotlib.pyplot library had been used to display results for both the modes using a *contourf* plot.

As mentioned above, a point on a plot displays just one position of the machine. The position of the machine expressed by point P in Figure 4 is shown under the plot. To get this position, the skidder was placed to the default position (front part facing toward 9 o'clock, on a contour line, as shown in Figure 3), then rotated (α = 0) by the angle of 150°, then the rear frame was articulated by the angle of 20°.

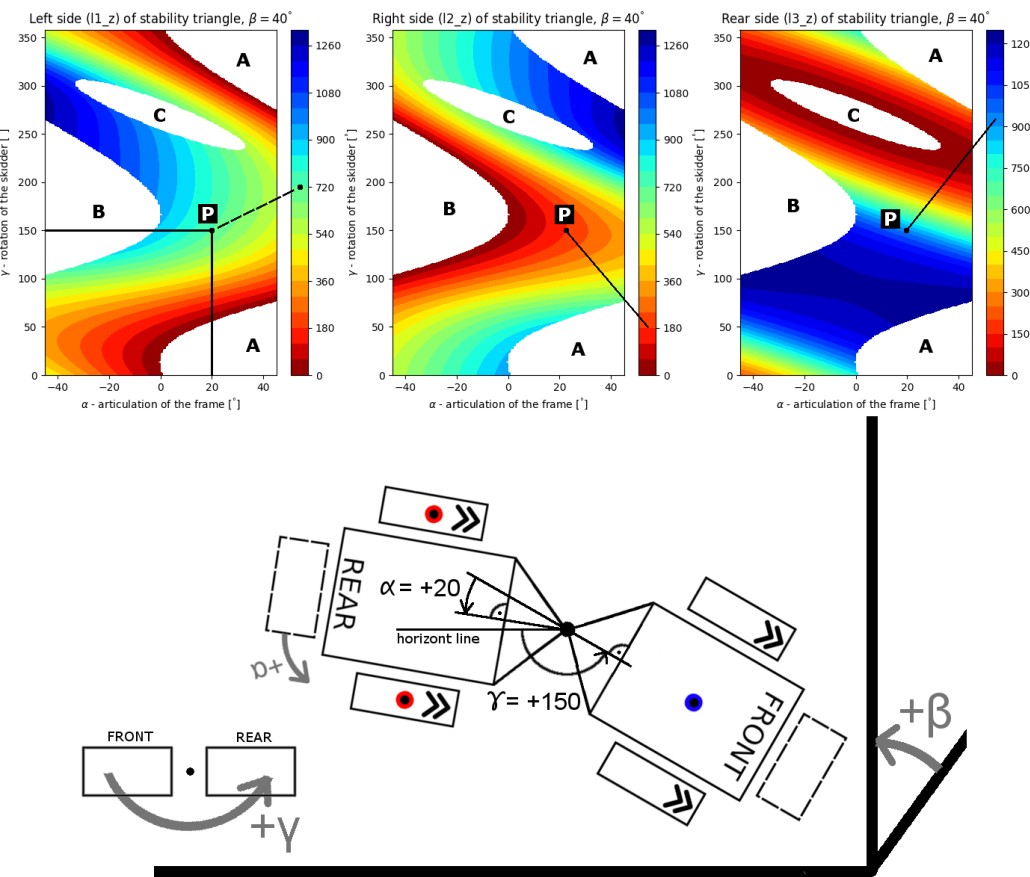

**Figure 4.** How to read the plots. Input parameters: $\alpha$—articulation of frame, $\gamma$—rotation of skidder, $\gamma$—slope angle (constant value). P—a combination of input parameters, A—area of instability, due to the left side (tipping line) of stability triangle, B—area of instability, due to right side of stability triangle, C—area of instability, due to the rear side of stability triangle. The first and second images display lateral stability, and the last one stands for longitudinal stability. Color bars show the distance of the gravity vector (for the whole machine, including additional loads) to tipping lines in [mm].

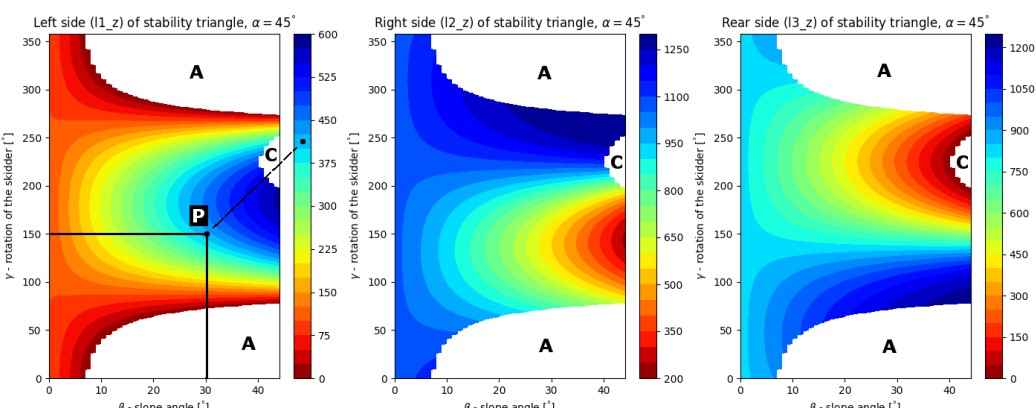

**Figure 5.** Example of plot in the second mode. Input parameters: $\alpha$—articulation of frame (constant value), $\gamma$—rotation of skidder, $\beta$—slope angle. P—combination of input parameters, A—area of instability, due to the left side (tipping line) of stability triangle, C—area of instability, due to the rear side of stability triangle. Color bars show the distance of the gravity vector (for the whole machine, including additional loads) to tipping lines in [mm].

## 3. Results

### 3.1. The Skidder with the Full Water Tank (C2 Configuration)

The stability of this configuration (Table 2) is expected to be the worst. The Figure 6 was added for a better understanding of the plots. The configuration is compared with the unloaded skidder (C1 configuration), as shown in Figure 7. The slope angle is 18°. Articulation of the frame lies in the interval $\langle -45°; 45° \rangle$, and rotation of the skidder lies in the interval $\langle 0°; 359° \rangle$. These intervals cover all the possible combinations of the skidder's positions on the slope. The same ranges we used in all of the plots below.

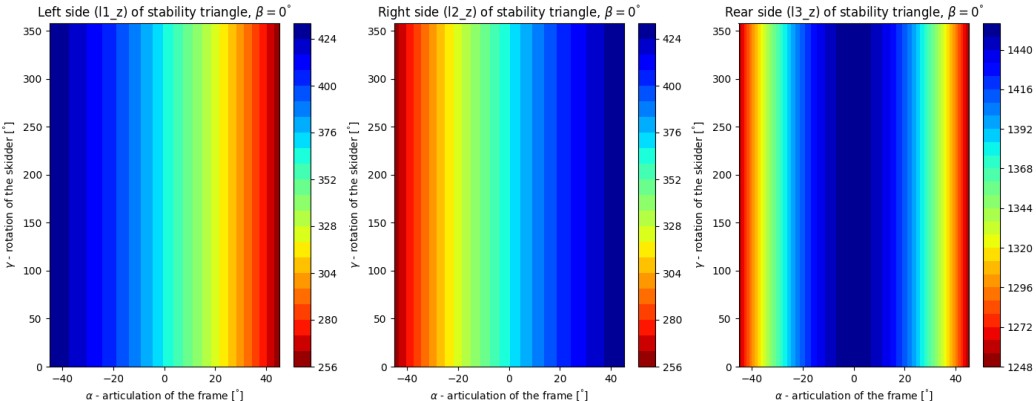

**Figure 6.** Stability plot for unloaded skidder (C1 configuration) on horizontal plane. Color bars show the distance of the gravity vector (for the whole machine) to tipping lines in [mm].

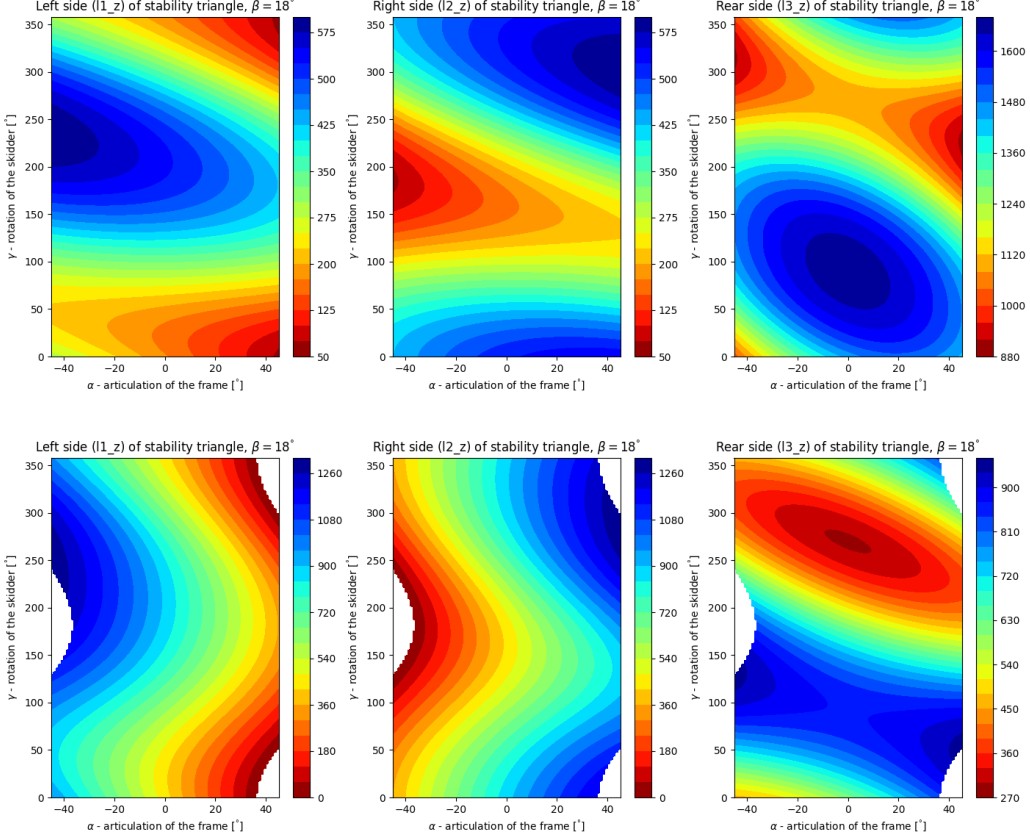

**Figure 7.** Comparison of firefighting devices stability. Upper plots are valid for an unloaded skidder (C1 config.), lower ones for C2 config.—full, 2000 L water tank behind the rear axle (RAW).

The unloaded skidder is stable in the range of the intervals above (the plots contain no white areas). This is, however, not valid for the skidder with the full water tank. It is unstable if frame articulation $\alpha$ is higher than $|-37°|$ and rotation of the skidder $\gamma$ is around $180°$ (left white area), or $\gamma$ is zero and $\alpha$ is more than $37°$ (right white areas). These positions on the slope represent the front half-frame on a contour line and the rear half-frame placed higher in the articulated positions. If the color bar's ranges for the left and right sides of the stability triangle are compared, the result is that the stability for the device equipped with the full water tank is higher, but within the limit above. It is an effect of the changed position of the center of gravity (CG), due to the heavy tank. The CG is moved toward the rear axle, and the distance between CG (point T, Figure 3) and FPP (Figure 3) points is higher, which results in higher distances of l1_z and l2_z, as shown in Figure 7. The comparison of color bars for the rear side of the stability triangle shows substantially decreased stability of the device equipped with the tank.

### 3.2. The Skidder with the Half-Filled Water Tank (C3 Configuration)

This configuration is an attempt to balance the utility value of the firefighting system and its slope stability. As mentioned above, the tank can be filled repeatedly. The result is shown in Figure 8. The stability of the system (in comparison with the full water tank) is better or worse depending on a criterion. If one compares color bars values for side tipping lines (l1_z, and l2_z) on Figures 7 (lower plots) and 8, the result is that the firefighting device with the full water tank is more stable, however, stability to rear tipping line is worse for the system with the full water tank. If the criterion includes also areas of instability, the system with the half-filled water tank provides visibly a little better results. However, such an adapter is not intended for motion on steep slopes in a contour line direction. If there is a road there, it should use it, or it should utilize the best stability of the system, that is to the rear tipping line. That means to follow the line in an upwards direction. It is almost doubled in comparison with the full tank. Then, the better configuration is the one with the half-filled water tank. It corresponds with terrain observations, where the main problem of the full water tank configuration (C2) was losing stability to the rear tipping line if a slope angle was too high. From a stability point of view, additional weight in the front of the skidder could allow for transporting more water in the tank.

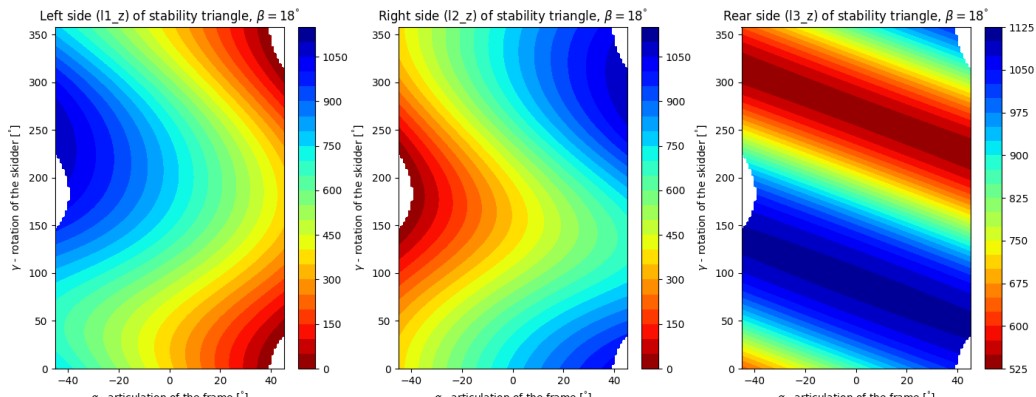

**Figure 8.** Firefighting device stability for half-filled water tank (1000 L) behind rear axle (C3 config.).

### 3.3. The Skidder with the Full Water Tank and Additional Weight in the Front (C4 Configuration)

The main reason for this configuration is to transport more water (full tank) and improve the stability when the motion of the firefighting device is considered upward, in the fall line direction. The additional weight in the FAW position is 500 kg. The stability plot is shown in Figure 9.

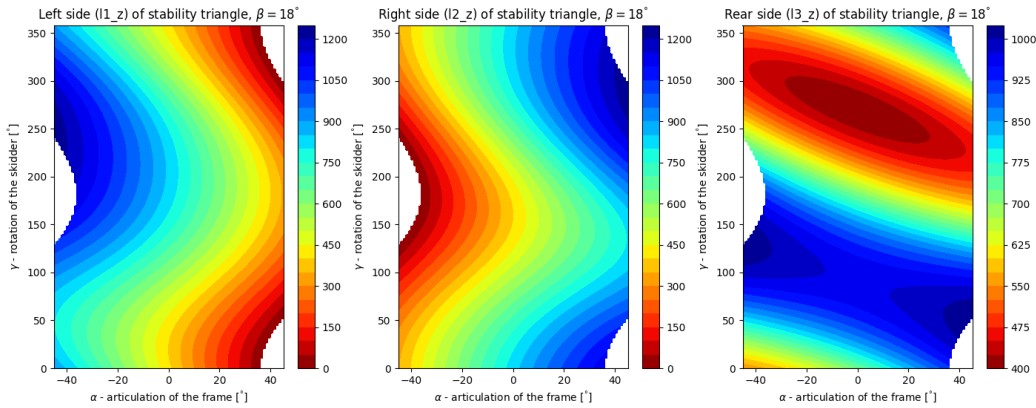

**Figure 9.** Firefighting device stability for full water tank (2000 L) behind rear axle, and additional weight (500 kg) in the front (C4 configuration).

The stability of the device in numbers is a combination of the full (C2) and half-filled (C3) water tank configurations. White areas of instability for the C4 configuration are almost the same as in the case of the full tank (C2 configuration), while lateral stability is worse. Longitudinal stability is similar to the stability for the C3 configuration. A more detailed comparison can be done with a point-by-point comparison only. This is why a code that displays a subtraction of two plots was written, as shown in Figure 10.

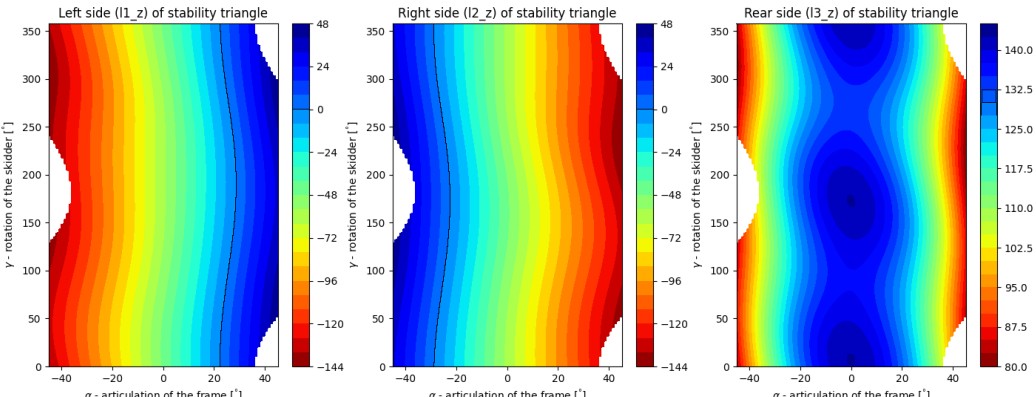

**Figure 10.** Comparison of stability for the half-filled tank (C3 config.) and full tank with additional front load (C4 config.). The plot displays the same values as the plots above, however, color bars show values computed as C3 minus C4 distances [mm] of gravity vector to tipping lines. Positive values mean that the C3 configuration is more stable than C4. Negative values mean that the C4 configuration is more stable than C3.

The comparison (Figure 10) can be evaluated using default position of the firefighting system ($\alpha = 0°$, $\gamma = 0°$). At this position, better lateral stability ensures configuration with additional front load (C4) (negative value in the plots). At $\alpha = 22°$, the lateral stability is the same for both configurations (left tipping line), while longitudinal stability (l3_z) is better for the configuration with the half-filled tank. The maximal difference in longitudinal stabilities (Figure 10) is 142.5 mm for non articulated frame ($\alpha = 0°$). That means that the use of ballast weight creates a significantly better result.

### 3.4. The Skidder with the Half-Filled Water Tank and Additional Weight in the Front (C5 Configuration)

This configuration (Figure 11) completes possible combinations of front-rear loads. In comparison with the configuration without the front load (C3), it provides similar results as the comparison with the full water tanks (C2 vs. C4), displayed in Figure 12. In general, an additional front load makes lateral stability worse, and longitudinal stability better,

and it can improve maneuverability by decreasing areas of instability. If we compare the stability of this configuration with the configuration without front load (C3) point-by-point, it is visible that the lateral stability of the device with front load is worse and longitudinal stability is better (Figure 13).

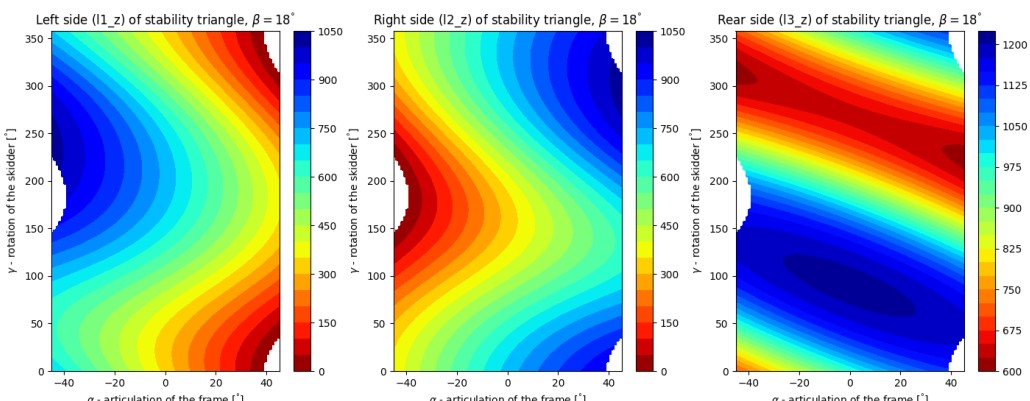

**Figure 11.** Firefighting device stability for half-filled water tank (1000 L) behind rear axle, and additional weight (500 kg) in the front (C5 config.).

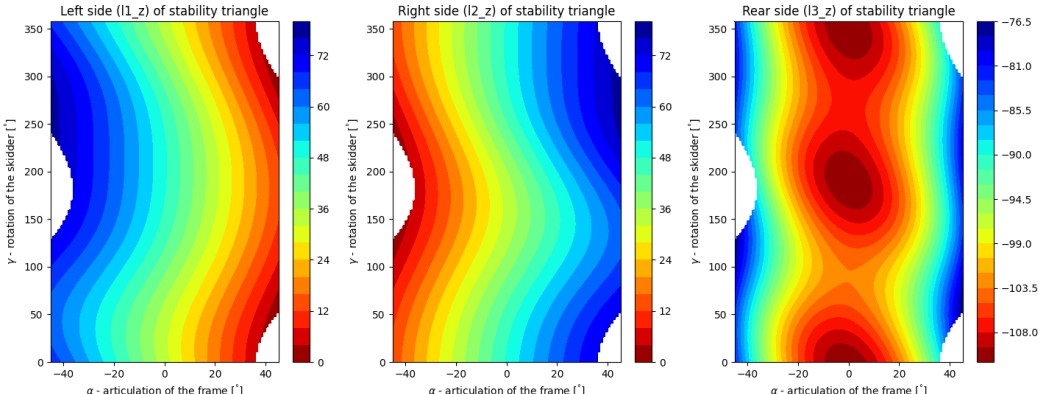

**Figure 12.** Comparison of stability for a full water tank (C2 config.) and the same tank with additional front load (C4 config.) The color bars show values computed as C2 minus C4 distances [mm] of the gravity vector to the tipping lines. Positive values mean that the C2 configuration is more stable than C4. Negative values mean that the C4 configuration is more stable than C2.

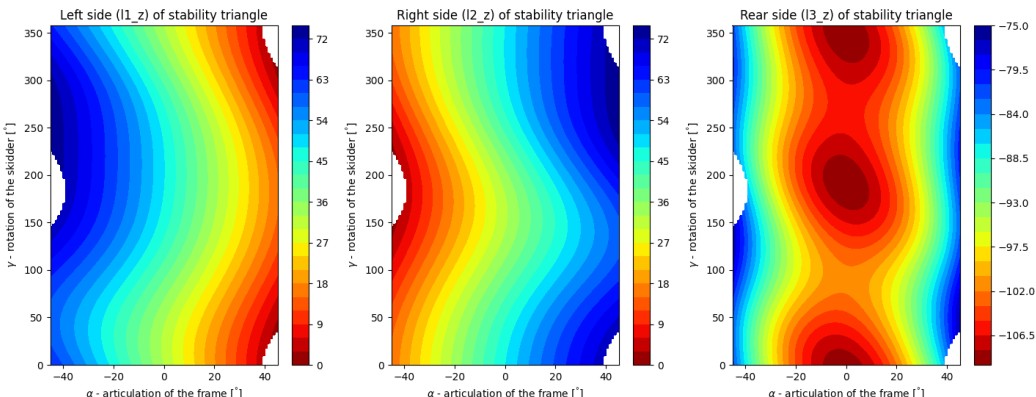

**Figure 13.** Comparison of stability for the half-filled tank (C3 config.) and the same tank with additional front load (C5 config.) Color bars show values computed as C3 minus C5 distances [mm] of gravity vector to tipping lines. Positive values mean that the C3 configuration is more stable than C5. Negative values mean that the C5 configuration is more stable than C3.

## 4. Discussion

The performed analyses of static stability utilized a simplified alternative skidder model, derived from the LKT-81 forestry skidder. The positions of FG and RG points were approximated and placed on the centerline. This allows for creating comparisons between configurations only. In the case of the use of the real skidder data, for the analyses above, the results could reflect the real stability of the real firefighting adapter. However, it requires the skidder's CAD (Autodesk Inventor, Solidworks, Catia, Creo Parametrics, etc.) model (that is available for the manufacturer only. It is not valid for the LKT-81, because it was designed before CAD era.), or physical measurements of the skidder. If the model is available, the positions of FG and RG points can be computed. Another possibility is to measure them. It is possible with lifting and tilting tests. Then, the center of gravity can be computed. In comparison with a non-articulated vehicle [24], the measurements may include articulated positions of the skidder.

The firefighting adapter based on the skidder with front axle oscillation and the articulated frame was investigated in five configurations/combinations of front and rear loads. The solution of determination of static stability herein described and applied, provides a complete display of stability for each configuration and range of variable parameters, using one plot per tipping line. The results showed, that a rear load, in form of the water tank, decreases longitudinal stability. However, it also improves lateral stability. That is a positive result. The negative one is that better lateral stability contains also areas of instability. Better longitudinal stability can be achieved with a front-load however it can decrease lateral stability. However, the firefighting skidder device is not intended for motion on steep slopes in a contour line direction. In general, there is a possibility to use a bigger tank and a bigger front ballast weight and equilibrate them to achieve the same result, as herein presented. That is the static solution. In the case of a motion in a real forest, a terrain micro-relief should be taken into account. It contains obstacles that can dynamically change an instant position of the adapter, and create dynamic forces. Future research could be focused on their influence on stability. Comparison plots then can display differences in stability, point-by-point.

If the stability and solution of the firefighting adapter are compared with other solutions (in references), the result has some benefits and disadvantages. The adapter is cheaper than a one-purpose vehicle and can be prepared and stored on a forest owner's site. In case of need, it can be employed and transported to a place accessible from a wildfire area using an ordinary forestry skidder, used every day in logging operations. The water tank then provides a renewable resource of water for firefighting operations in a place where the water is missing and needed. The disadvantages of the solution are worse stability and a smaller volume of the tank. Both disadvantages are partially equilibrated with the high terrain accessibility of a forestry skidder. The water tank in a right place can improve and simplify the liquidation of forest fires. It is possible to imagine that the tank can also be filled with water by unmanned aerial vehicles. However, future perspectives are based on future firefighting ways.

## 5. Conclusions

The globally warmer climate emphasizes a need to combat wildfires. Regardless of a firefighting strategy, the availability and amount of extinguishing agents are important for a successful fight against wildfires. The firefighting adapter provides a relatively cheap way to place a water source in a place accessible from a wildfire area. However, the stability of such a solution is important. The performed research shows that such a solution is stable from a longitudinal point of view. The lateral stability is also maintained, except for maximal articulations of the frames. Some positions of the machine with the tank on the slope also result in better lateral stability, in comparison with the machine without the tank. However, these positive or negative findings regarding lateral stability are less important than longitudinal stability. The use of front additional weight improves the longitudinal stability but increases overall weight and thus the dynamical stability will

be probably decreased. The positions of centers of gravities of the skidder's front and rear frames were approximated, and an older forestry skidder was used. Future research can use a new type of skidder, with known precise positions of the centers of gravities. This allows for investigating the stability of the real skidder in a specific configuration. The herein investigated solution of the adapter is joined with a specific forestry skidder. This supposes a forest owner to own the firefighting adapter that is usable with available forestry skidders. Future research can be widened to more universal solutions. This will enable the movement of the ownership of the adapter from a forest owner to a fire brigade.

**Author Contributions:** Conceptualization, J.M.; methodology, J.M.; validation, J.M., R.H. and M.H.; formal analysis, J.M.; investigation, J.M., R.H. and M.H.; resources, J.M., R.H. and M.H.; data curation, J.M.; writing—original draft preparation, J.M.; software, J.M.; writing—review and editing, J.M.; visualization, J.M. and R.H.; supervision, J.M. and R.H. All authors have read and agreed to the published version of the manuscript.

**Funding:** This research was funded by "Operational Programme Integrated Infrastructure" (contract ITMS 313011T720).

**Institutional Review Board Statement:** Not applicable.

**Informed Consent Statement:** Not applicable.

**Data Availability Statement:** Not applicable.

**Conflicts of Interest:** The authors declare no conflict of interest.

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
