# Peer review of "Research on Static Stability of Firefighting Adapter"

_forests, doi:10.3390/f13081180_

Round 1

Reviewer 1 Report

The submitted manuscript for review is an innovative approach to research/modeling the mobility/stability of cable-skidder primarily produced for timber extraction, which has been purposefully modified to firefighting.

In the chapter on materials and methods, the co-authors should explain the method of determining the coordinates of centers of gravities for front and rear skidder frames, on the values of which the presented analyzes of static stability of the vehicle are based.

The manuscript does not contain chapter – Discussion.

I am not a native English speaker, but many sentences are written in active form, which should be written in passive form.

Author Response

We thank the reviewer for the comments concerning our manuscript. We hope the manuscript has been corrected according to the reviewer´s comments.

Point 1:

In the chapter on materials and methods, the co-authors should explain the method of determining the coordinates of centers of gravities for front and rear skidder frames, on the values of which the presented analyzes of static stability of the vehicle are based.

Response 1:

We did not measure it. We analyzed a simplified alternative skidder model with similar parameters only. The model has the same dimensions, overall weight. The weight distribution is similar, and the positions of centers of gravities are approximated, and rounded. The reason and limits are described in rows 70-80.

Point 2:

The manuscript does not contain chapter – Discussion.

Response 2:

We added this paragraph.

Point 3:

I am not a native English speaker, but many sentences are written in active form, which should be written in passive form.

Response 3:

We hope we corrected it successfully.

Note:

The original manuscript contained an error - the position of RAW point. It also has been corrected.

Reviewer 2 Report

The manuscript in this form requires serious major revisions and repairs.

The paper is not an independent scientific paper and the introduction is unclear and incomprehensible without prior reading of the scientific paper under number 2 in the references.

The text of the paper is not written in the scientific style of writing (english passive form) and requires major improvements to the text and mandatory proofreading of English.

The introduction needs to be completely reorganized.  The development and application of firefighting skidder as well as the purpose and objectives of the research need to be better explained.

Citations are insufficient or completely incorrect.

Examples:

Line 26: "There are many studies" (4) - (14) - totally wrong, it is necessary to list individual citations (results) from previous research.

Line 29 and 30: "using a generally known physical principle" - citations are missing.

The first sentence of the Introduction should be moved to Acknowledgmet or Funding.

The idea of ​​the research is very interesting, research method is well explained, the results are well presented but there is a lack of explanations and scientific discussions in the presentation of the results. There is a lack of discussion, although the chapter is entitled "Results and discussion". Complete the chapter with discussion and relevant references to similar research.

The results of the research could be much clearer and more understandable presented by data processing in more specialized softwares (Simulation X, MathLab etc)

For a better understanding of the results and the possibility of wider application of the results, it is necessary to explain the position of the points FG and RG and the procedure of their measurement and determination in the real articulated forest machines. As a rule, these points are not exactly the centerline of the vehicle or part of the vehicle, but have a certain offset (in mm) from the centerline.

Author Response

We thank the reviewer for the comments concerning our manuscript. We hope the manuscript has been corrected according to the reviewer´s comments.

Point 1:

The manuscript in this form requires serious major revisions and repairs. The paper is not an independent scientific paper and the introduction is unclear and incomprehensible without prior reading of the scientific paper under number 2 in the references.

Response 1:

We modified a bigger part of the manuscript. We also added better description of the adapter, so the previous reference under number 2 is not necessary for the purpose of the research in the manuscript.

Point 2:

The text of the paper is not written in the scientific style of writing (english passive form) and requires major improvements to the text and mandatory proofreading of English.

Response 2:

We hope we corrected it successfully.

Point:3

The introduction needs to be completely reorganized.  The development and application of firefighting skidder as well as the purpose and objectives of the research need to be better explained.

Response 3:

We rewrote the entire paragraph.

Point 4:

Citations are insufficient or completely incorrect.

Examples:

Line 26: "There are many studies" (4) - (14) - totally wrong, it is necessary to list individual citations (results) from previous research.

Line 29 and 30: "using a generally known physical principle" - citations are missing.

Response 4:

We modified the citations. Now, they contain only relevant items. We described them in Introduction (rows 37-62).

Point 5:

The first sentence of the Introduction should be moved to Acknowledgmet or Funding.

Response 5:

We rewrote the entire paragraph.

Point 6:

The idea of ​​the research is very interesting, research method is well explained, the results are well presented but there is a lack of explanations and scientific discussions in the presentation of the results. There is a lack of discussion, although the chapter is entitled "Results and discussion". Complete the chapter with discussion and relevant references to similar research.

Response 6:

We added Discussion paragraph.

Point 7:

The results of the research could be much clearer and more understandable presented by data processing in more specialized softwares (Simulation X, MathLab etc)

Response 7:

Our research is based on the simulation using Python that is very similar to Matlab. We agree that tools like Simulation X could enhanced the research, however their require more data. Authors of the paper have experiences with MSC.Adams, Project Chrono and virtual reality. We should like to use such a tool if a new version of the firefighting adapter will be designed.

Point 8:

For a better understanding of the results and the possibility of wider application of the results, it is necessary to explain the position of the points FG and RG and the procedure of their measurement and determination in the real articulated forest machines. As a rule, these points are not exactly the centerline of the vehicle or part of the vehicle, but have a certain offset (in mm) from the centerline.

Response 8:

We did not measure positions of FG and RG points. We analyzed a simplified alternative skidder model with similar parameters only. The model has the same dimensions and overall weight. The weight distribution is similar, and the positions of centers of gravities are approximated, and rounded. The reason and limits are described in rows 70-80. We also added a short description at rows 240-249.

Reviewer 3 Report

General comment:

The authors of this article describe the static stability of a water fire skidder device tank behind the rear axle. This remains an important and original research topic, I appreciate the authors' work. The research is reasonably interesting, although a couple of questions need to be reviewed and addressed to increase the scientific value of this research.

Title: the authors could slightly modify the title giving a more scientific approach, I suggest "Evaluation of the static stability of the fire-fighting skidder device”

References: Bibliographic references are too poor; I believe authors should adding more references.

Abstract

Abstract requires revision to improve understanding of the work. Some sentences need to be rephrased for clarity. Furthermore, authors should give a more scientific slant to make the reader understand in a few lines: the state of the art; the purpose of their work; the materials and methods used; the results achieved. At the moment the setting is not clear I suggest rewriting it.

Keyword: Words from the title should not be used as keywords. I suggest to modify, reducing the number and the means

Introduction

The introduction does not provide a clear and adequate state of the art of what is discussed in this document, it is too poor. There is no adequate description with scientific references on other contextual works. I suggest broadening the bibliographic search and increasing the contents of the introduction.

Figure 1: Is this figure an example or is it a photo taken during data collection? there is no reference in the text of this figure.

Figures: Figure captions are sometimes really long. Is it possible to reduce and move the text in the paragraph?

Materials and Methods

Material and method are partially organized as regards the description of the scientific methodology to be applied. However, it is not at all clear what type of methodological protocol was applied.

I advise the authors to rewrite the materials and methods by clearly describing the methodological approach of this research. For example the description of the skidder and the study area, the paths taken by the machine.

Line 39: We decided..... the third person is favored throughout the scientific manuscript

Result and Discussion

The results do not present any statistical approach, I believe that the authors should implement the manuscript with an analysis that gives a statistical value

The discussions need to be compared with other scientific studies of the sector. Some parts require careful revision to clarify, extend and verify some inconsistencies.

But the description of the materials and methods is much more exhaustive than the Results and Discussion paragraph.

Conclusions

The conclusion appears to be a discussion of the results that does not clearly describe the contribution of this research and potential future work.

Author Response

We thank the reviewer for the comments concerning our manuscript. We hope the manuscript has been corrected according to the reviewer´s comments.

The authors of this article describe the static stability of a water fire skidder device tank behind the rear axle. This remains an important and original research topic, I appreciate the authors' work. The research is reasonably interesting, although a couple of questions need to be reviewed and addressed to increase the scientific value of this research.

Point 1:

Title: the authors could slightly modify the title giving a more scientific approach, I suggest "Evaluation of the static stability of the fire-fighting skidder device”

Response 1:

We changed the title of the article.

Point 2:

References: Bibliographic references are too poor; I believe authors should adding more references.

Response 2:

We hope we corrected it successfully.

Point 3: 

Abstract

Abstract requires revision to improve understanding of the work. Some sentences need to be rephrased for clarity. Furthermore, authors should give a more scientific slant to make the reader understand in a few lines: the state of the art; the purpose of their work; the materials and methods used; the results achieved. At the moment the setting is not clear I suggest rewriting it.

Keyword: Words from the title should not be used as keywords. I suggest to modify, reducing the number and the means

Response 3:

We changed substantial part of the manuscript in all the paragraphs. We also modified the keywords.

Point 4:

Introduction

The introduction does not provide a clear and adequate state of the art of what is discussed in this document, it is too poor. There is no adequate description with scientific references on other contextual works. I suggest broadening the bibliographic search and increasing the contents of the introduction.

Figure 1: Is this figure an example or is it a photo taken during data collection? there is no reference in the text of this figure.

Figures: Figure captions are sometimes really long. Is it possible to reduce and move the text in the paragraph?

Response 4:

We rewrote the entire paragraph. We modified the caption of Fig.1 and Fig.3, which contained the same items as the caption of Fig.2. We also modified the caption of Fig.4 – a part of the caption we moved into the main text of the article.

Point 5:

Materials and Methods

Material and method are partially organized as regards the description of the scientific methodology to be applied. However, it is not at all clear what type of methodological protocol was applied.

I advise the authors to rewrite the materials and methods by clearly describing the methodological approach of this research. For example the description of the skidder and the study area, the paths taken by the machine.

Line 39: We decided..... the third person is favored throughout the scientific manuscript

Response 5:

We rewrote the paragraph to emphasize that our research is based on computations and visualization of the result. We hope that all the grammar issues were corrected successfully.

Point 6:

Result and Discussion

The results do not present any statistical approach, I believe that the authors should implement the manuscript with an analysis that gives a statistical value

Response 6:

We are not sure if we understand well the reviewer’s comment. We used no data recorded in a real environment. Out results are based on computer computations/simulations that provide exact results, that need no statistical processing.

Point 7:

The discussions need to be compared with other scientific studies of the sector. Some parts require careful revision to clarify, extend and verify some inconsistencies.

Response 7:

We rewrote the paragraph. We agree that such a comparison could be valuable for the discussion, but the firefighting adapter is a new solution (rows 19-26), and there is no a similar scientific study, suitable for the comparison. We hope we identified all the parts that require revision, and corrected them successfully. We also corrected the position of RAW point.

Point 8:

But the description of the materials and methods is much more exhaustive than the Results and Discussion paragraph.

Response 8:

We modified Results and Discussion paragraphs.

 Point 9:

Conclusions

The conclusion appears to be a discussion of the results that does not clearly describe the contribution of this research and potential future work.

Response 9:

We removed this paragraph.

Round 2

Reviewer 3 Report

General comment:

The authors of this article describe the static stability of a water fire skidder device tank behind the rear axle. This remains an important and original research topic, I appreciate the authors' work. The research is reasonably interesting, and has been improved, in my opinion it still needs some improvements.

References: Bibliographic references are still too scarce, there is no bibliographic comparison with other similar works. I believe that authors should look for further bibliographical references.

Introduction

The introduction has been improved, but needs further references and a clear research purpose.

Line: 15-46-53…. it would be advisable that the numbers of the bibliographic references are in ascending order (1-2-3-4-5 ...) and not random.

Line: 63 I suggest the authors to specify the goal of this study "the purpose of this study is"

Result and Discussion

The discussions need to be further compared with other similar scientific studies.

Conclusions

I suggest to the authors to insert a final conclusion that can give a key to reading your research and above all future perspectives.
